# Alzheimer’s Disease Pathogenesis: Role of Autophagy and Mitophagy Focusing in Microglia

**DOI:** 10.3390/ijms22073330

**Published:** 2021-03-24

**Authors:** Mehdi Eshraghi, Aida Adlimoghaddam, Amir Mahmoodzadeh, Farzaneh Sharifzad, Hamed Yasavoli-Sharahi, Shahrokh Lorzadeh, Benedict C. Albensi, Saeid Ghavami

**Affiliations:** 1Center for Motor Neuron Biology and Disease, Columbia University, New York, NY 10032, USA; eshraghi.mehdi@gmail.com; 2Department of Pathology and Cell Biology, Columbia University, New York, NY 10032, USA; 3St. Boniface Hospital Albrechtsen Research Centre, Division of Neurodegenerative Disorders, Winnipeg, MB R2H2A6, Canada; umadlimo@gmail.com (A.A.); balbensi@sbrc.ca (B.C.A.); 4Medical Biology Research Center, Health Technology Institute, Kermanshah University of Medical Sciences, Kermanshah 6734667149, Iran; amir.biochem@gmail.com; 5Department of Cellular and Molecular Medicine, Faculty of Medicine, University of Ottawa, Ottawa, ON K1H 8M5, Canada; fsharifzad@gmail.com (F.S.); yasavoli.h@gmail.com (H.Y.-S.); 6Department of Human Anatomy and Cell Science, Rady Faculty of Health Sciences, Max Rady College of Medicine, University of Manitoba, Winnipeg, MB R3E 0J9, Canada; lorzadeh.sh@gmail.com; 7Department of Pharmacology & Therapeutics, University of Manitoba, Winnipeg, MB R3T 2N2, Canada; 8Research Institute of Oncology and Hematology, Cancer Care Manitoba-University of Manitoba, Winnipeg, MB R3E 0V9, Canada; 9Biology of Breathing Theme, Children Hospital Research Institute of Manitoba, University of Manitoba, Winnipeg, MB R3E 0V9, Canada; 10Faculty of Medicine, Katowice School of Technology, 40-555 Katowice, Poland

**Keywords:** mitochondria, inflammation, autophagy, mitophagy, microglia, Alzheimer’s, neurodegeneration, neuroinflammation

## Abstract

Alzheimer’s disease (AD) is a debilitating neurological disorder, and currently, there is no cure for it. Several pathologic alterations have been described in the brain of AD patients, but the ultimate causative mechanisms of AD are still elusive. The classic hallmarks of AD, including amyloid plaques (Aβ) and tau tangles (tau), are the most studied features of AD. Unfortunately, all the efforts targeting these pathologies have failed to show the desired efficacy in AD patients so far. Neuroinflammation and impaired autophagy are two other main known pathologies in AD. It has been reported that these pathologies exist in AD brain long before the emergence of any clinical manifestation of AD. Microglia are the main inflammatory cells in the brain and are considered by many researchers as the next hope for finding a viable therapeutic target in AD. Interestingly, it appears that the autophagy and mitophagy are also changed in these cells in AD. Inside the cells, autophagy and inflammation interact in a bidirectional manner. In the current review, we briefly discussed an overview on autophagy and mitophagy in AD and then provided a comprehensive discussion on the role of these pathways in microglia and their involvement in AD pathogenesis.

## 1. Introduction

Alzheimer’s disease (AD) is a progressive neurodegenerative condition characterized by loss of memory and abnormal mood and behavior. Most cases of AD are sporadic and aging is the biggest risk factor for AD [1]. The disease was first described by Dr. Alois Alzheimer in 1906; however, the fine structure of the pathologic hallmarks of AD (i.e., Aβ plaques and neurofibrillary tangles) were described by using electron microscopy in 1963 [2,3]. In recent decades, many therapeutic approaches that targeted Aβ and tau pathologies have failed to show the desirable efficacy for alleviating cognitive impairments in AD patients [4,5]. Possible reasons for these failures may include: treating the patients during later stages of AD, targeting the wrong pathologic mechanism, and an insufficient understanding of the complex multifactorial nature of AD, which may require combination therapy over mono-therapy [6].

However, recent studies suggest a critical role for microglia and neuroinflammation in AD [7]. While under normal conditions, microglia’s main function is to support the integrity and survival of neuronal networks, in AD a special group of microglia (i.e., disease associated microglia or DAM) appear that seems to contribute to the aberrant neuroinflammation in the brain and consequent neurodegeneration [8,9]. The focus of the current review is to survey the current knowledge about the role of microglia and neuroinflammation in AD. Specifically, we would like to emphasize the importance of a known but underrated pathology in AD, which is impaired autophagy and mitophagy. Impaired autophagy and mitophagy are present in the vast majority of AD cases [10]. It is believed that Aβ/tau aggregation and impaired autophagy/mitophagy and neuroinflammation are intertwined processes and create several vicious cycles in the course of AD. In this review paper, first, we will discuss the current knowledge of impaired autophagy/mitophagy and neuroinflammation in AD. Next, we will provide evidence of the interplays between impaired autophagy/mitophagy and microglial over-activation/neuroinflammation in AD.

## 2. Alzheimer’s Disease; A Global Health Crisis

Dementia has been classified as a neurocognitive disorder which presents with cognitive and/or behavioral impairments and are accompanied with a progressive need for support with activities of daily living (ADL) [11]. Dementia is an imminent global health crisis; approximately 50 million people are living with dementia worldwide [12,13,14]. However, due to the rise of seniors among the global population and lack of an effective treatment, the number of individuals with dementia is growing significantly. This number is expected to increase considerably, reaching 152 million by 2050, a 204% increase [14,15]. According to the World Health Organization (WHO), the global cost associated with healthcare related to dementia was 1% of the aggregated global gross domestic product (GDP), or 604 and 818 billion USD, in 2010 and 2015, respectively [16,17,18]. Rising numbers of people with dementia will place greater demands on both formal care (e.g., residential facility, medication, hospitalization) and informal care (e.g., family members, friends, or neighbors) [19]. Therefore, dementia has a great effect not only on individuals with the disease, but also on their families and friends.

AD is the most common form of dementia; it accounts for 60–80% of all dementia cases [20,21,22]. According to the WHO, AD was the fifth leading cause of global death in 2016 [23]. Currently, there is no cure for AD; however, some pharmacological therapeutics can delay the progression of the disease and/or alleviate some symptoms of AD [24,25,26]. Typically, physicians recommend various options for AD treatment based on the severity of the disease. The AD severity is categorized into mild, moderate, and severe [27]. The mild stage of AD usually exists for two to four years [28]. The moderate stage of AD can last from two to 10 years [29]. And, the severe stage of AD typically usually lasts for one to three years. In this final stage, physical and cognitive skills are severely affected; therefore, patients need extensive support with daily living and communication [30].

Overall, more than 90% of AD cases belong to sporadic category which is suggested to be caused by complex genetic and environmental factors [31,32]. Several risk factors are involved in the progression of AD [33,34,35,36]. Aging is the greatest risk factor for the prevalence of AD [20,37]. Another risk factor is family history of the disease [38]. There are three genes identified for familial AD, including *Amyloid Beta Precursor Protein* (*APP*) [39], *Presenilin 1* (*PSEN1*), and *Presenilin 2* (*PSEN2),* but only one gene, *Apolipoprotein E* allele *ε4* (*APOEε4*), is associated with the sporadic forms of AD [40,41]. Other genetic factors, such as mutations in *Triggering Receptor Expressed On Myeloid Cells 2* (*TREM2*), are also linked with high risk of late-onset AD [42]. Sex is another risk factor for AD [43]; females experience significant decreases in sex hormones starting at menopause, which increases the risk of AD [44].

Histologically, AD is characterized by amyloid plaques, neurofibrillary tangles (NFTs), neuronal loss, and gliosis [28,45,46]. Aβ is generated from the cleavage of APP by β-secretase (BACE1) and γ-secretase enzymes, Aβ monomers aggregate into oligomer forms of Aβ first and ultimately to amyloid plaques in the brain [47]. Current theories suggest that Aβ peptides accumulate for more than a decade prior to the appearance of clinical symptoms in AD [48]. NFTs, a crucial AD pathology, are insoluble twisted fibers, or the so-called tangles that are found inside neurons. The tangles consist mostly of a protein called tau, which is a component of microtubule organization [49]. One of the functions of microtubules is to maintain an internal transport network for the trafficking of vesicles and other cargo inside the cells. In AD, tau protein is hyperphosphorylated, which leads to compromised microtubules, and hence, disruption of several cellular processes, such as proliferation, differentiation, protein trafficking, and cellular morphology [50] (Figure 1). Although the accumulation of NFTs is correlated with the neuronal loss and cognitive decline, it is not certain if NFTs are the actual cause of neurodegeneration in AD [51,52].

## 3. Autophagy and AD

### 3.1. Autophagy

Autophagy (meaning “self-eating” in Greek) is one of the main pathways for the removal of damaged proteins and organelles. It works by carrying the protein aggregates and damaged organelles into lysosomes [53,54,55,56]. However, the function of autophagy in neurons is more than just simple housekeeping tasks and involves several neuronal specific functions such as axonal guidance, synaptic transmission, proper neuronal connectivity, and neural stem cell development [57,58]. In fact, the impairment of autophagy is suggested to be a major contributing factor to the brain dysfunction in neurodegeneration [10,59].

Autophagy is categorized into three major forms: microautophagy, chaperone-mediated autophagy (CMA), and macroautophagy. These different forms of autophagy are discriminated based on the way that the targeted cargo is transported to lysosomes for degradation [60,61,62]. Macroautophagy (hereafter autophagy) is the major form of autophagy in the cells [63]. Autophagy is also induced by starvation [64]. In this way, autophagy is important for cells by providing them with essential nutrients [65,66].

The formation of autophagosome involves several steps, including the induction of autophagy, nucleation and the formation of autophagy membranes (i.e., phagophore), and expansion of the membrane to form a double membrane vesicle (i.e., autophagosome) [60,67,68]. It has been shown that in yeast, this process is highly regulated by a set of “Autophagy Related Genes” (ATGs) [69,70,71]. ATGs are evolutionary conserved genes and their paralogues exist in higher species including mammals. Within the cells, the products of these genes interact and form multi-molecular complexes, which are responsible for the regulation of different steps of autophagy [72]. These complexes in mammalian cells include the ULK1/2-ATG13-FIP200 complex, the PtdIn3K complex, and the ubiquitin-like conjugation complex Atg12-Atg5-ATG16L [73,74,75] (Figure 2).

The ULK1/2-ATG13-FIP200 complex plays a pivotal role in the induction of autophagy [76]. This complex consists of unc-51-like autophagy-activating kinase 1 (ULK1, the mammalian homologue of yeast Atg1), ATG13, ATG101a, and FIP200 (AKA RB1CC1) (Figure 2A). Under physiologic conditions, this complex is inhibited by mTORC1 complex. During starvation, mTORC1 complex is inhibited and ULK1/2-ATG13-FIP200 complex initiates the autophagy process within cells. In addition, starvation and energy depletion activates 5′-adenosine monophosphate activated protein kinase (AMPK), which results in the phosphorylation of ULK1 and nucleation of ULK1/2-ATG13-FIP200 complex [77,78] (Figure 2A). It is believed that mitochondria-associated endoplasmic reticulum membranes (MAMs) are the origin of autophagy membranes [79,80].

The ULK1/2-ATG13-FIP200 complex activates Beclin1 complex (consisting of ATG14L-Beclin1-hVps34-p150). Upon activation, both complexes translocate to the site of nucleation of autophagy membrane, phagophore (Figure 2B). As a result, membrane lipid phosphatidylinositol is phosphorylated, generating phosphatidylinositol 3-phosphate (PtdIns(3)P) on the surface of the phagophores. PtdIns(3)P is recognized by proteins such as WIPI2, resulting in the recruitment of ATG16L to phagophores (Figure 2E). This results in activation of two ubiquitin-like conjugation complex involved in autophagy. First, ATG12 is conjugated to ATG5 by the function of ATG7 and ATG10. The ATG12/ATG5 conjugate then makes a complex with ATG16L [81] (Figure 2C). Then, the ATG12/ATG5/ATG16L complex binds to and activates ATG3, resulting in the conjugation of a second complex; ATG8/LC3 and phosphatidylethanolamine (PE) on the surface of autophagosomes [82] (Figure 2D). It seems that these two ubiquitin-like conjugation complexes are interconnected, and together mediate the formation of autophagosomes and navigate their fusion with lysosomes [83].

### 3.2. Dysregulation of Autophagy in AD

There is solid evidence that autophagy flux is impaired in AD [84]. In cells, the formation of different types of autophagy vacuoles (AVs) follows a distinct order [85]. First, autophagosomes are formed by the extension of phagophore membranes. Then, amphisomes are formed by fusion of autophagosomes with endosomes. Finally, autolysosomes are made through fusion of amphisomes or autophagosomes with lysosomes [86,87]. One of the striking ultrastructural changes in AD is the accumulation of all forms of AVs inside affected neurons [88,89]. Indeed, the magnitude of AVs accumulation in AD is so high that it resembles some primary lysosome storage disorders (LSDs) [90] (Figure 3). Multilamellar bodies (MLBs) are membrane-bound cellular organelles, which accumulate inside the cells under pathologic conditions, such as LSDs, and represent impaired cholesterol and lipid metabolism [91]. Accumulation of MLBs is also reported within dystrophic neurites in AD [88]. It is noteworthy to mention that some LSDs are accompanied with hallmarks of AD (e.g., NFTs and upregulated amyloidogenic process); however, this association happens during the very early stages of life [92].

Impaired autophagy was also confirmed in laboratory models of AD. Mutations in *PS1* genes are among the genetic causes of AD. Using fibroblasts derived from AD patients, it was shown that mutations of the *PS1* gene results in disruption of lysosomal acidification/proteolysis [92]. In one interesting study, it was observed that depletion of PS1 in neurons derived from induced pluripotent stem cells (iPSCs) impaired the formation of autophagosomes. The authors also found that the expression of several autophagy-related genes was diminished in these cells mediated through ERK/CREB signaling pathway in a γ-secretase-independent manner [93]. Autophagy is also impaired in other mouse models of AD; mice overexpressing mutant APP show disrupted autophagy, probably due to the toxic effect of the β-secretase-cleaved carboxyl-terminal fragment (βCTF) of APP on lysosomes [94].

It has been shown that mutations in ATG genes can recapitulate neurodegenerative phenotypes in mice. Conditional knock outs of *Atg5* and *Atg7* genes in mouse models resulted in accumulation of polyubiquitinated proteins as cytoplasmic inclusion bodies and eventually death of neurons [95,96]. On the other hand, the enhancement of autophagy in mouse models of neurodegenerative disorders (for example through inhibition of mTORC1) reduces the number of cytoplasmic inclusion bodies and improves the phenotype in these animals [97,98]. In addition, the enhancement of autophagy alleviated memory loss in several animal models of AD paralleled by lower levels of Aβ_42_, Aβ_40_ and hyperphosphorylated Tau in brain tissue [84,94]. Lysosomal biogenesis is regulated by transcription factor EB (TFEB). Overexpression of TFEB in mouse models of AD also reduced the number of protein deposits and alleviated behavioral abnormalities [99] [100].

Impaired autophagy is also connected to production and accumulation of Aβ in AD. Yu et al. reported that isolated AVs from brain tissues of AD mice are enriched in APP, Aβ peptides, and gamma secretase complexes [101]. Another study showed that inhibition of autophagy flux in healthy neurons results in accumulation of APP enriched AVs in perikarya and neurites [101,102,103] *Beclin1*, the mammalian orthologue of yeast *Atg6*, plays a pivotal role in autophagy through facilitating lipid membrane extension, cargo recruitment, and autophagosome maturation [104]. It has been shown that the levels of *Beclin1* is decreased in the affected brain areas of patients with AD during early stages of the disease. Additionally, overexpression of *Beclin1* in a mouse model of AD diminished both intracellular and extracellular amyloid aggregation [105].

## 4. Mitophagy and AD

### 4.1. Mitophagy

Mitochondria are the main source of energy in neurons and their proper function is crucial for synaptic transmission, calcium homeostasis, redox signaling, synaptic plasticity, and neuronal survival. [106,107]. As the power engines of the cells, all the components of mitochondria are prone to severe oxidative damage [67]. Damaged mitochondria contribute to cell toxicity and death through generation of excessive amounts of reactive oxygen species (ROS) [108]. Therefore, the presence of a powerful mitochondrial quality control (MQC) system is pivotal for cells to screen and neutralize such harmful events quickly. In cells, the MQC system includes several mechanisms including those involving mitochondrial fission and fusion, mitochondrial unfolded protein responses (mUPR), and mitophagy [109,110] Mitochondria possess very dynamic biogenesis processes that respond to major damage by generating new and healthy mitochondria that fuse to the existing mitochondria, or by segregation of the unrecoverable mitochondria, which are degraded through mitophagy [111,112,113].

Mitophagy, the selective form of autophagy for mitochondria, is a crucial process for neuronal health and survival (summarized in Figure 4). Impaired mitophagy has been implicated in the pathogenesis of several neurodegenerative disorders, such as AD and Parkinson’s disease (PD) [114]. The molecular mechanisms underlying selective autophagy of mitochondria has not been fully deciphered. Several studies have suggested that sustained depolarization of the mitochondrial inner membrane stabilizes the PTEN-induced kinase 1 (PINK1) on the outer mitochondrial membrane (OMM) (Figure 4). Consequently, phosphorylation of mitofusin2 (Mfn2) and ubiquitin molecules by PINK1 results in recruitment of Parkin, an E3 ubiquitin ligase, to the OMM. Finally, several proteins on damaged mitochondria are ubiquitinated by Parkin and become recognizable for autophagy receptor proteins, such as optineurin (OPTN), NDP52, p62, and NBR [115]. Recent studies discovered more mitophagy receptors, including Autophagy and Beclin1 Regulator 1 (AMBRA1), BCL2 Interacting Protein 3 Like (BNIP3L/Nix), and FUN14 Domain Containing 1 (FUNDC1). AMBRA1 can induce mitophagy via binding to LC3 in both Parkin-dependent and -independent manners [116]. Nix (AKA BNIP3L) was first recognized as a mitophagy receptor in erythroid cells; however, it has been shown that Nix can induce mitophagy in a variety of other cell types including neurons [117]. FUNDC1 is a mitophagy receptor protein and induces mitophagy under hypoxic conditions [118] (Figure 4). After targeting the damaged mitochondria, mitophagy receptors covalently bind to the nascent autophagosomes’ proteins, LC3 and GABARAP, and form protein bridges between the phagophore membrane and the OMM. This leads to the elongation (mediated by LC3) and closure (mediated by GABARAP) of the phagophore membrane and results in complete engulfment of mitochondrion [119] (Figure 2G). Fusion of lysosomes with autophagosomes is the last stage of mitophagy, which happens through the binding of PLEKHM1 and HOPS (on the phagosome) with Rab7 (on the lysosome) (Figure 2H). Finally, the engulfed mitochondrion breaks down inside the lysosome by the function of hydrolase enzymes [120] (Figure 2I).

*BCL2 like 13* (*BCL2L13*, also known as *BCL-Rambo*) is a *BCL2* like gene. It contains all four BH domains and a single transmembrane domain. *BCL2L13* was first considered as a pro-apoptotic gene. [121,122]. However, Murakawa et al. showed that this gene is indeed a mammalian orthologue to yeast *ATG32* and induces mitophagy independent of the Parkin pathway. The authors showed that BCL2L13 binds to LC3-II leading towards engulfment of mitochondria within an autophagosome [123]. *BCL2L13* is constitutively expressed in the human fetus and promotes embryonic development [124]. Additionally, it has been shown that *BCL2L13* has a role in the regulation of energy metabolism and fat homeostasis [125].

### 4.2. Dysfunction of Mitochondria in AD

Accumulating evidence have revealed that mitochondrial dysfunctions appear very early in the course of AD (even before the appearance of Aβ plaques and NFTs) [126,127,128,129]. These include downregulation of the enzymes involved in the tricarboxylic acid cycle (TCA) cycle, reduced oxidative phosphorylation and increased ROS production [130,131]. PET scan studies on AD patients have shown that glucose uptake is decreased in several brain regions, which is known as the “hypometabolism state of AD brain,” which is probably due to large-scale mitochondrial dysfunction [132]. The level of PGC-1-alpha, the master regulator of mitochondrial biogenesis, is reduced in brain samples of AD patients, suggesting that mitochondrial biogenesis is also impaired in AD [133,134]. In addition, several studies on brain samples from AD patients confirmed accumulation of impaired mitochondria in neurons within the affected brain regions. Indeed, it seems that the clearance of damaged mitochondria (e.g., through mitophagy) is severely compromised in AD [114,135].

A body of evidence supports the role of Aβ/tau pathology in mitochondrial dysfunction in AD. Overexpression of APP in cultured neurons results in extensive abnormal function and defective biogenesis of mitochondria [136]. Aβ and p-tau interact directly with several mitochondrial components (e.g., VDAC1 and complex IV) and interfere with their functions [137,138,139]. Several studies on samples from AD patients and mouse models also revealed that the mitochondrial permeability transition pore (mPTP) is dysregulated in AD, resulting in various types of mitochondrial dysfunction, such as decreased mitochondrial transmembrane potential, decreased mitochondrial respiration rates, increased ROS production, and mitochondrial swelling in AD brain [140,141,142].

It was shown that in a cell line model of AD, Aβ oligomers (AβOs) induce mitochondrial fragmentation accompanied by reduced mitofusin1 and 2 (Mfn1/2) levels. It has also been reported that excessive generation of ROS by AβOs activates the Cyclin-dependent kinase 5 (Cdk5) pathway, which in turn results in decreased expression of Mfn1/2 [143]. Additional work revealed that AβOs mediate activation of dynamin-related protein 1 (Drp1), the fundamental component of mitochondrial fission machinery, resulting in enhanced fragmented mitochondria and subsequent cell death in AD [144].

Increased cytoplasmic calcium is detrimental for neurons and might result in neuronal death [145]. Endoplasmic reticulum (ER) and mitochondria are the main reservoirs of intracellular calcium and play a crucial role in calcium homeostasis inside cells [146,147]. It has been shown that Aβos disrupt the intracellular calcium balance, and hence, impair the normal function of mitochondria [148]. Previous reports also demonstrated that Aβ enhances glutamatergic neurotransmission via *N*-methyl-D-aspartate receptors (NMDARs) and increases tau phosphorylation, which might initiate neuronal cell death via mitochondrial pathways [149,150] (Figure 3).

Tau pathology also contributes to the mitochondrial dysfunction in AD [151,152,153,154,155,156,157,158]. Several groups reported that p-tau results in downregulation of *Optic Dominant Atrophy 1* (Opa1) and *Drp1*, and upregulation of *Mfn1/2* [151,159,160,161]. Additionally, p-tau blocks mitophagy by inhibiting the recruitment of Parkin to damaged mitochondria (Figure 4) [162]. Hyperphosphorylation of tau also induces alterations in the binding ability of tau to microtubules and leads to formation of paired helical filament (PHF) and disruption of axonal transport of mitochondria in AD [163,164].

Does mitochondria dysfunction contribute to the pathogenesis of AD or is it merely a consequence of other pathologic events, such as Aβ or p-tau aggregation? Mutations in mitochondrial fusion genes have been shown to result in mitochondrial fragmentation and AD pathologies [165]. In addition, inhibition of mitochondria complex I (via administration of rotenone) induced tau pathology in rat brain [166] [167]. Additionally, it has been shown that oxidative stress increases the activity of the γ-secretase complex, where this phenomenon is mediated through post-translational modifications of one of its components, nicastrin by 4-hydroxynonenal [168]. Similarly, excessive ROS contributes to p-tau aggregation through phospholipid peroxidation [169,170]. On the other hand, selective depletion of mitochondrial cytochrome c oxidase (COX) in neurons alleviated amyloid pathology in a mouse model of AD [171]. Thus, it seems that while Aβ and p-tau aggregation aggravate mitochondrial dysfunction, mitochondrial dysfunction also enhances Tau phosphorylation and Aβ aggregation. Indeed, there might be a vicious cycle between mitochondria dysfunction and classic AD pathologies [167,172].

### 4.3. Dysfunction of Mitophagy in AD

It is believed that mitophagy is severely compromised in AD (Figure 3) [173,174]. Impaired mitophagy results in the accumulation of damaged mitochondria within neurons and contributes to the neurodegenerative process [84,118,175] (Figure 3). Recent advancements in ultrasensitive next generation sequencing (NGS) disclosed the unusual accumulation of mitochondria DNA (mtDNA) mutations within the brain of patients with AD, which is believed to be due to accumulation of damaged mitochondria in neurons [176,177]. Additionally, base excision repair (BER) and DNA double-strand break repair (DSBR) are two major DNA repair pathways in neurons, which are downregulated in AD [178,179,180,181]. Interestingly, *Ataxia Telangiectasia Mutated* (*ATM*), a DNA repair gene downregulated in AD, plays an important role in autophagy and mitophagy [182,183]. However, the precise relationship between defective DNA repair and mitophagy/autophagy impairment is not clear yet and needs further studies to determine the exact nature of causality.

*Sirtuin1* (*SIRT1*) and *Sirtuin3* (*SIRT3*) are two genes with neuroprotective characteristics. Reductions in SIRT1 and SIRT3 activities have been reported in neurodegenerative conditions, including AD [184]. It seems that SIRT1 executes its neuroprotective function by induction of autophagy/mitophagy via deacetylation of autophagy proteins, such as ATG5, ATG7, and ATG8/LC3. SIRT1 also stabilizes PINK1 and upregulates mitophagy proteins including LC3 and Nix/BNIP3L [185,186]. Therefore, disruptions of SIRT1 function result in suppression of mitophagy and subsequent accumulation of damaged mitochondria in neurons [186]. SIRT3 is an activator of FOXO3; it has been shown that the function of FOXO3 is essential for autophagic flux in neurons [187].

Neural NAD^+^ deficiency has also been suggested as another potential reason for mitophagy impairment in AD. The proper homeostasis of intracellular ATP and NAD^+^ levels is crucial for neuronal function and survival. Optimum levels of NAD^+^ affect neuronal health by maintaining a fine balance between mitophagy and mitochondrial biogenesis. It has been shown that decreased level of NAD^+^ triggers the aggregation of misfolded proteins that compromises mitophagy and ultimately results in neuronal death [188,189,190].

## 5. Microglia, Neuroinflammation, and AD

### 5.1. Neuroinflammation in AD

Neuroinflammation has been implicated to play a substantial role in AD [191,192]. It has been shown that activation of microglia precedes Aβ and tau pathologies within the brain of AD patients and animal models [193,194]. Additionally, increased levels of inflammatory mediators (e.g., IL-1β) have been consistently reported in AD brain [195]. Inflammasomes are multi-protein complexes that play pivotal roles in inflammatory pathways inside the cells. They become activated upon exposure of cells to pathogen-associated molecular patterns (PAMP) or danger-associated molecular patterns (DAMP) and promotes the cleavage of Caspase-1 and the release of inflammatory cytokines such as IL-1β and IL-18 [196]. In the brain, NLRP3 and NLRP1 (two sensor components of inflammasome complexes) are expressed in neurons and microglia [197]. It has been reported that both NLRP3 and NLRP1 inflammasomes are over-activated in AD [198,199,200]. Halle et al. reported that Aβ is a potent activator of inflammasomes in microglia The authors showed that upon phagocytosis by microglia, Aβ induces lysosomal damage and leakage of Cathepsin B into cytosol resulting in activation of inflammasomes [199].

Thus far, the efficacy of several anti-inflammatory therapeutics has been tested for AD treatment (https://www.alzforum.org/therapeutics/timeline/inflammation, accessed on 10 March 2021) [200]. We believe that the success of the anti-inflammatory approach highly depends on further discovery of molecular events, which specifically mediate over-activation of microglia in AD brain. For this mean, after providing a brief background about microglia, we will discuss how microglia is over-activated in AD by providing current evidence about the role of its surface receptors. Then, we will review current knowledge about the role of several mediators of inflammation, which are known to target and activate microglia in AD.

### 5.2. Microglia

As a part of the innate immune system, microglia play an important role in maintaining brain homeostasis in the central nervous system (CNS) [201,202]. These “macrophage-like” cells account for virtually 10% of the cell population in a healthy adult brain [203]. Microglia react rapidly to the changes in their microenvironments and play important roles in CNS homeostasis under normal and pathologic conditions, including tissue damage and infection [204,205].

Initial microscopic examinations of the brain implied that under normal conditions, microglia remain in their spatial coordinates for a long time and show no obvious activity. Thus, the term “resting microglia” was coined to describe them at this stage. However, it has now been established that the “resting microglia,” indeed, monitor their surrounding environments very actively through constant remodeling of their processes [206]. Additionally, microglia provide support for the neurons in some parts of the brain by secretion of neurotrophic factors such as insulin-like growth factors (IGF) [207]. As a matter of fact, frequent functional and structural interactions between neurons and microglia directly influence several fundamental events in the CNS, such as neural plasticity and brain development [208].

### 5.3. Over-Activation of Microglia in AD

#### 5.3.1. Complement Receptors

It has been shown that the expression levels of complement receptors in microglia are increased in AD [209,210]. Upregulation of complement receptors in AD seems to contribute to Aβ plaque formation, Aβ phagocytosis, and Aβ-mediated inflammation [211]. Interestingly, activation of microglial complement receptors in AD is associated with an upregulation of the nuclear factor kappa B (NF-κB) signaling pathway [212]. This finding suggests that NF-κB-dependent microglia activation might be a crucial contributor to AD. Crehan et al. reported that blockage of complement receptors in AD inhibits microglial activation and leads to neuronal protection [213]. Furthermore, silencing the complement receptors rescued tau pathology and reversed tau-mediated inflammation in AD brain [214]. Additionally, depletion of complement receptors prevents AD-associated dysfunctional phagocytosis, which suggests a role for the complement system in AD etiology [215].

#### 5.3.2. Fc Receptors

It has been shown that microglial FcRs interact with Aβ peptides, suggesting a role for FcRs in AD pathogenesis [216]. Microglial FcR-mediated phagocytosis triggers the expression of cytokine genes, leading to inflammatory cascades. FcRs activation in primary human microglia stimulates certain immunoglobulins and induces macrophage inflammatory protein (MIP)-1α through activation of the NF-κB pathway [217,218]. In accordance with this, depletion of microglial FcRs suppresses tau pathology and neuroinflammation in AD [216]. Additionally, administration of antibodies against FcRs could lower tau absorption rates by microglia [219]. Several lines of evidence revealed that microglial FcRs are also involved in Aβ phagocytosis [220,221]. Antibodies against Aβ peptide triggered AD microglia to clear Aβ plaques through FcR-mediated phagocytosis. This finding suggests that the anti-Aβ antibody increases microglial-mediated clearance of Aβ in an Fc-dependent phagocytosis manner [216,222]. However, some groups reported that Fc binding is not required for clearance of Aβ by microglia [223,224].

#### 5.3.3. Scavenger Receptors

A recent analysis of brain samples from APP23 transgenic mice showed high levels of Scavenger Receptor A-1 (SCARA-1) in microglia surrounding Aβ plaques [225]. Additionally, microglia derived from SCARA-1 knockout mice displayed a significant reduction in Aβ phagocytosis by microglia [226]. These findings suggest that SCARA-1 is involved in Aβ clearance by microglia.

Class B of scavenger receptors, SCARB-2 (AKA CD36), is expressed by microglia in AD brains [227]. Binding of CD36 to Aβ mediates activation of microglia to produce ROS, proinflammatory cytokines, and chemokines associated with AD [228,229]. Upon activation, CD36 forms heterodimers with TLR-4 and TLR-6 on microglia, resulting in an increase of production of ROS and IL-β [230]. Khoury et al. reported that microglia derived from CD36 knockout mice produce fewer amounts of pro-inflammatory cytokines such as TNFα and IL-1β upon exposure to Aβ [228]. Additional evidence showed that CD36 increases the expression of NF-κB subunits significantly in cells expressing TLR4-TLR6 [230]. These findings suggest that CD36 plays a critical role in Aβ-induced ROS production in AD microglia. In accordance with these findings, injection of Aβ into CD36 knock-out mouse brains induced less accumulation of microglia, suggesting a significant role of CD36 in mediating inflammation in response to Aβ stimulation [228].

#### 5.3.4. Receptor for Advanced Glycation Endproducts (RAGE)

Several lines of evidence have reported that inflammatory processes found in AD are linked to microglia activation by Aβ interaction with RAGE [231,232,233]. RAGE is a member of the immunoglobulin (Ig) superfamily of cell surface molecules with high homology to the neural cell adhesion molecules (N-CAM) and MUC18 [234]. Thus far, several ligands have been identified for RAGE with important roles in diabetes, inflammation, tumors, and neurodegeneration [235]. RAGE has a large (344 amino acid long) extracellular domain and a single transmembrane domain. Using atomic force microscopy, Chaney et al. showed that soluble RAGE binds to soluble Aβ and inhibits aggregation of Aβ peptides. On the other hand, interaction of Aβ with membrane bound RAGE elaborates inflammatory response [236].

The expression of RAGE is increased in AD brain [237]. In AD brain, Aβ interacts with RAGE, which activates NF-κB signaling pathways in microglia and enhances the production of inflammatory mediators and ROS [238,239]. Deane et al. showed that blocking RAGE in an AD mouse model reduced RAGE-Aβ interactions and decreased oxidative damage [240]. On the other hand, microglial overexpression of RAGE in a transgenic-AD mouse model increased the production of proinflammatory mediators such as IL-1β, IL-6, and TNF-α after Aβ treatment [238].

Interestingly, it has been reported that binding of Aβ to RAGE mediates Aβ endocytosis. Park et al. reported that knockdown of RAGE prevented accumulation of intracellular Aβ and the subsequent cellular toxicities. Interestingly, inhibition of p38 MAPK prevented internalization of Aβ peptides, suggesting that p38 acts downstream of RAGE for endocytosis of Aβ peptides [241].

Aβ also binds to RAGE on the surface of neurons and induces the expression and release of macrophage-colony stimulating factor (M-CSF) by neurons. M-CSF is a strong activator of microglia represented by upregulation of scavenger receptors and ApoE, and production of excessive ROS in these cells [236].

#### 5.3.5. Triggering Receptor Expressed on Myeloid Cells 2 (TREM2)

TREM2 is a microglial receptor highly expressed in AD brains [242,243]. Depletion of TREM2 impairs Aβ phagocytosis by microglia, implying that TREM2-Aβ interaction may be crucial for clearance of amyloid plaques in AD brain [244]. Another study showed that a deficiency of TREM2 decreases Aβ-associated microglia in an AD mouse model [245]. However, a recent study showed that overexpression of TREM2 did not alter the amount of amyloid plaque in AD brain [246]. Earlier studies suggested that loss of TREM2 function mitigates neuroinflammation, which protects brain atrophy in the context of tauopathy [247]. In one study, *TREM2* gene knockout in AD mice resulted in reduced microglial activity and tau pathology [247], while in another study *TREM2* deletion resulted in the elevated tau phosphorylation and aggregation [248]. These seemingly contradictory results on Aβ and tau load in TREM2^−/−^ AD mouse models might be due to the type or age of the mouse models being used [249,250,251] and mandate more investigations to address the role of TREM2 in Aβ pathology.

#### 5.3.6. CD33

CD33 is a membrane receptor expressed on myeloid cells. CD33 expression is increased in microglial cells in AD brain [252]. CD33 seems to interfere with Aβ clearance by microglia [253]. Indeed, high CD33 levels in AD brains were associated with lower microglial clearance of Aβ, while a deletion of CD33 in AD mice attenuated Aβ pathology [252]. Interestingly, microglial TREM2 and CD33 display reverse effects on microglial activity as well as Aβ pathology in AD mice [252]. It has been shown that CD33 knockout increases microglial activation and production of inflammatory mediators in AD mice, while TREM2 knockout has opposite effects. It seems that TREM2 acts as a downstream target of CD33; however, the exact nature of crosstalk between CD33 and TREM2 in the context of AD needs more investigation [252].

#### 5.3.7. Toll-like Receptors (TLRs)

A high level of TLRs has been detected in brain samples of AD mouse models. Moreover, mutations in TLR genes displayed reduced microglial activity and increased Aβ levels in the brain of AD mouse models [254]. Microglia derived from TLR knock-out mice showed a significant reduction in the production of ROS upon exposure to Aβ, which links TLR to the oxidative response induced by Aβ [253]. There is additional evidence suggesting TLRs trigger various signaling pathways in microglia that lead to the production of proinflammatory molecules and clearance of Aβ [255]. Microglia derived from TLR knockdown mice displayed reduced expression of TNF-α, IL-1β, and IL-6 [256]. TLR knockdown mice also showed decreased expression levels of proinflammatory molecules after injection of Aβ into their brain cortex [256]. These findings suggest that inhibition of TLRs in AD brain could delay AD progression by decreasing neuroinflammation and improving Aβ clearance. The expression of TLRs has been revealed to be associated with an increase in the level of tau in AD brains [257]. These findings designate that TLR signaling is associated with tau pathology [258]. Interestingly, a major signaling target of the TLRs is NF-κB, which induces inflammatory cytokines such as IL-1β, IL-6, and IL-8. Several lines of evidence report that TLR-mediated NF-κB activation is conserved throughout the evolution of phylogenetically different species [259,260,261].

### 5.4. Increased Levels of Proinflammatory Mediators in AD

#### 5.4.1. Apolipoprotein E (APOE4)

APOE4 is mainly expressed by microglia in the brain [262]. APOE4 increases the risk of AD through deficient Aβ and tau clearance by impairing microglial phagocytosis [263,264,265]. APOE4 has been also shown to impact Aβ clearance by the blood–brain barrier (BBB) and decreases Aβ plaque load [266]. Additional evidence in a mouse model of tauopathy demonstrated that APOE affects tau pathology, independent of Aβ pathology [267]. In AD brain, microglial APOE provides a protective Aβ- and tau-induced inflammatory response such as TNFα and IL6 [267,268]. The expression and activity of APOE4 is elevated after brain injury [269]. Several studies show that APOE protects BBB integrity through NF-κB signaling in AD [270,271]. Another study found that upregulation of APOE by Aβ in human astroglia is regulated by the NF-κB signaling pathway [272]. Moreover, all APOE isoforms bind to TREM2 in AD [273]. APOE-TREM2 binding results in the transcriptional shift of AD microglia from a homeostatic phenotype to a neurodegenerative phenotype [262,274].

#### 5.4.2. Chemokines

Both in vitro and in vivo studies reported that microglial chemokine ligand1 (CX3CL1; fractalkine) also plays a critical role in AD pathology [275]. CX3CL1 is likely to alter the microglial state to a more neuroprotective one by acting on the microglial chemokine receptor 1 (CX3CR1) in microglia. Impairment of CX3CL1-CX3CR1 results in microglial dysfunction [276]. The expression of CX3CL1 is elevated in AD brain [277]. CX3CL1-CX3CR1 plays a critical neuroprotective role in brain by reducing inflammatory mediators (e.g., TNF-α, IL-1β, IL-6), neurotoxicity, and microglial activity [278]. Silencing CX3CR1 in AD mouse models displayed increased expression of inflammatory markers and enhanced AD pathology [279,280]. Additional evidence shows that deletion of microglial CX3CR1 elevated the activity of the NF-κB signaling pathway, which results in elevated levels of chemokines such as IL-1β [281].

The CXCL12/CXCR4 pathway is among the most important neuroprotective pathways in CNS. CXCR4 is expressed by all cell types in the brain and plays several roles during the development of neurons [282]. Its natural ligand, CXCL12, is upregulated by pro-inflammatory mediators such as IL-1, TNF-α and LPS. CXCL12 also has an important role in the maintenance and recovery of neurons. For example, CXCL12 signaling promotes neuronal repair after brain damage by recruiting neural stem cells to the site of injury [283,284]. It has been reported that the activity of the CXCR4/CXCL12 pathway is severely diminished in AD. Additionally, inhibition of this pathway in mice resulted in cognitive deficits in this animal. On the other hand, increased levels of CXCL12 were shown to be protective against the neurotoxic effects of Aβ in mouse brain [285].

#### 5.4.3. High Mobility Group Box 1 (HMGB1)

Alarmins are endogenous equivalents of PAMPS, which are released following non-programmed cell death and recruit and activate receptor expressing cells of the innate immune system [286]. High mobility group box 1 (HMGB1) and S100 proteins are endogenous molecules that meet all the criteria to be classified as alarmins and play important roles in neuroinflammation [287].

HMGB1 resides primarily in the nucleus and has a role in chromatin architecture [288]. However, HMGB1 is released to the extracellular spaces under conditions such as infection, injury, inflammation, and non-programmed cell death. HMGB1 can be also secreted actively by activated myeloid cells. It has been shown that inflammasomes are the key regulators of active release of HMGB1 [289]. HMGB1 signaling is indeed at the cross section of sterile and pathogen induced inflammation. On its target cells, HMGB1 can bind to TLR2, TLR4, and RAGE and activate inflammatory responses [290]. Recently, Volchuk et al. reported that HMGB1 is only released under conditions that cause cell lysis (e.g., pyroptosis) [291].

The levels of HMGB1 are increased in AD, which is believed to contribute further to neuroinflammation in this disease [292,293]. It is also reported that HMGB1 binds to Aβ within extracellular space and reduces its clearance by microglia [294].

It has been suggested that HMGB1 can serve as a viable therapeutic target in several conditions related to neuroinflammation [295]. In an interesting experiment, Fujita et al. showed that injection of an antibody against HMGB1 inhibits neurite degeneration and recovers cognitive impairment in a mouse model of AD [296].

#### 5.4.4. S100 Proteins

S100 proteins (aka calgranulins) are a family of 25 calcium-binding proteins, which are released by activated myeloid cells to induce inflammation. S100 proteins have several intracellular and extracellular functions. Inside the cells, S100 proteins are involved in biologic functions, such as cell proliferation, cytoskeletal kinetics, and calcium homeostasis. S100 proteins are also secreted into the extracellular space, and indeed, act as DAMP molecules and stimulate immune cells by binding to receptors like RAGE and activate proinflammatory pathways such as NF-κB signaling [297]. S100 proteins play important roles in the brain homeostasis and several of them are reported to be involved in AD pathogenesis (including S100B, S100A1, S100A6, S100A7, S100A8, S100A9, and S100A12) [297]. Among them, S100 β is specifically expressed in the brain and is strongly associated with Aβ pathology in AD.

S100B is mainly produced by astrocytes in the brain. Its expression is regulated by cytokines such as TNF-α. At low expression levels (i.e., nanomolar concentrations), S100B acts as a neurotrophic and neuroprotective cytokine and protects neurons against neurotoxic cytokines and compounds such as TNF-α and glutamate [298]. On the other hand, higher concentrations of S100B (e.g., the ones reported in pathologic conditions such as AD and HIV infection) are detrimental to neurons through acute stimulation of RAGE [299]. Indeed, serum concentration of S100B increases drastically in AD patients and correlates with the severity of the disease [300].

It has been reported that high concentrations of S100B induce neuronal apoptosis through over-activation of ERK1/2, resulting in the excessive production of ROS and NOS in neurons [301,302]. At high concentrations, S100B also acts as a DAMP and activates microglia through binding to RAGE. This interaction mediates the activation of several parallel proinflammatory pathways in microglia including Ras/Rac1/NF-κB and Cdc42/Rac1/JNK/AP-1 and consequent expression and release of IL-1β, TNF-α, and COX-2 [302,303].

S100A8 and S100A9 are expressed by neurons and microglia in CNS and the levels of both peptides are significantly increased in AD [304,305]. S100A8/A9 heteromer is a potent proinflammatory cytokine and activates microglia through binding to TLRs [306]. Treatment of neuroblastoma cells with recombinant S100A8 increased Aβ42 production by these cells [307]. Additionally, exposure of microglia derived from AD patients to aggregated Aβ increased the expression of S100A8 in these cells [308]. It has been reported that S100A9 is able to mimic the behavior of Aβ peptides in the formation of polymeric structures that resemble amyloid architectures such as oligomers and fibrils [309]. Nasal administration of these forms of S100A9 resulted in impaired memory and learning in mice [310]. Co-aggregation of S100A9 with Aβ is also reported in AD. Post-mortem examinations have revealed the accumulation of S100A9 within amyloid plaques of sporadic and familial AD brains [311]. Interestingly, depletion of S100A9 in Tg2576 mice alleviated the cognitive deficits and Aβ aggregation in these mice [312].

## 6. Autophagy and Neuroinflammation in AD

Autophagy and inflammation are two biologic processes that are upregulated in cells in response to several stressful events. Interestingly, the two pathways interact at several points and modulate each other in a bidirectional manner [313]. In general, it is shown that inflammation induces autophagy inside the cells. Th1 cytokines (including IFN-γ, TNF-α, IL-1, IL-2, IL-6, and TGF-β) are proinflammatory cytokines, while Th2 cytokines (e.g., IL-4, IL-10 and IL-13) suppress inflammation [314]. It has been reported that Th1 cytokines activate autophagy, while Th2 cytokines inhibit autophagy in the cells [315]. On the other hand, autophagy itself is a regulator of inflammation inside the cells. It has been reported that autophagy decreases the secretion of proinflammatory cytokines, including IL-1β, IL-6, and TNF-α through the degradation of several components of inflammatory machinery in the cells [316]. In this way, cells seem to use autophagy as negative feedback to modulate the cytotoxic effects of inflammation. Therefore, impaired autophagy is implicated in sustained inflammation in tissues and might contribute to pathogenesis of chronic inflammatory conditions [313].

### 6.1. Role of Impaired Autophagy in Neuroinflammation

During neurodegenerative conditions, an excessive amount of misfolded protein aggregates leads to microglial activation and neuroinflammation. It has been shown that induction of autophagy in general is effective in the mitigation of this inflammatory condition [317]. Specifically, induction of autophagy in microglia ameliorates neuroinflammation due to degradation of misfolded proteins, damaged mitochondria (i.e., mitophagy), and activated inflammasomes [318].

It has been proposed that prolonged exposure to Aβ impairs microglial autophagy in AD [319]. This hypothesis was suggested by evidence from AD experimental models and patients that show dysregulation of autophagy in microglia [320]. It has been reported that the formation of autolysosomes is diminished within microglia in AD due to impaired fusion of autophagosomes with lysosomes [320,321]. Furthermore, the integrity of the lysosomal membrane is compromised due to the Aβ exposure. In an innovative experiment, Stoka et al. utilized Cathepsin D tracing and showed that the structure of the microglial lysosome is preserved after a short (2-h) exposure to Aβ [322]. However, long exposure (24 h) to Aβ resulted in lysosome permeabilization, indicated by extra lysosomal Cathepsin D [322]. In addition, neuroinflammation is another factor contributing to the impaired autophagy in the cells exposed to Aβ peptides [320]. It was shown that the inflammasome-forming NLRP3 and the pro-inflammatory cytokine TNFα are involved in impaired autophagy in microglia isolated from AD animal models [323,324].

Several studies showed that enhancing autophagy in neurons is protective against accumulation of misfolded and damaged proteins in AD brain [59,94,325]. Specifically, mounting evidence highlight a pivotal link between autophagy and microglial function in the nervous system (for detailed explanation please see Figure 5) [148,317,326]. Microglia are the main player of the Aβ-induced immune response in AD and contribute to neurodegeneration and synaptic dysfunction in this disease [327,328]. Microglia are also involved in phagocytosis of a wide range of cellular debris, such as apoptotic bodies, axonal fragments, and Aβ and p-tau deposits [7,329]. It has been shown that there is a close crosstalk between autophagy and phagocytosis in microglia [330]. Phagocytosis shares many molecular pathways with autophagy. It is an evolutionarily conserved mechanism in which recognition, engulfment, and lysosomal degradation of extracellular materials occurs by immune cells such as microglia and macrophages [329,331].

Mediators of autophagy orchestrate phagocytosis in microglia in various ways (for details, review Figure 5). For example, LC3 is associated with phagocytosis, a process known as LC3-associated phagocytosis (LAP) [332]. During this cellular event, LC3 recruitment to the single-membrane phagosomes is mediated by the activity of *ATG5* and *ATG7* [333] (Figure 5). Thereafter, Beclin1 and LC3 proteins translocate to the phagosome’s membrane, allowing the fusion of phagosome with lysosome (Figure 5). This process induces rapid acidification and degradation of the ingested organisms [334]. The cooperation of autophagy mediators with phagocytosis enhances the efficiency of microglial phagocytosis in the elimination of extracellular pathogens [335]. In a very recent study, Heckmann et al. showed that enhancement of LC3-associated endocytosis (LANDO), a distinct but similar pathway to LAP, facilitates the clearance of Aβ and alleviates neurodegeneration in murine models of AD [120]. They showed that LANDO is essential for recycling of Aβ receptors (e.g., TREM2) in microglia, and requires the function of *Rubicon*, *Beclin1*, *ATG5*, and *ATG7* genes (Figure 5). The authors also reported that this process protects against Aβ-mediated neuronal loss and memory impairment in a murine model of AD [120].

Accumulation of apoptotic bodies has been reported in AD [336]. Surprisingly, inhibition of autophagy in *Drosophila* brains via inhibition of ATG1 and mTORC1 complex, does not lead to neurodegeneration [336]. Thus, it seems that the removal of apoptotic cell bodies by microglia occurs through the activation of LAP and phagocytosis. It can be conferred that LAP may prevent neurodegeneration, even in the absence of autophagy flux [335].

Microglial Beclin1 acts as a regulator of Aβ removal and phagocytosis in AD [331]. Interestingly, the level of Beclin1 is significantly reduced in microglia isolated from human AD brains [331]. In microglia, Beclin1 is involved in the regulation of phagocytosis through regulating the expression of microglia surface receptors TREM2 and CD36 [331]. Additionally, it has been shown that Beclin1-driven autophagy modulates the response of Nre3 inflammasomes [337]. As mentioned before, neuroinflammation is an early hallmark in the course of AD. In a recent study, Houtman et al. reported that depletion of Beclin1 may affect cytokine production by microglia. Based on their study, there was a significant increase in the production of proinflammatory cytokines IL-1β and IL-18 by the activated microglia lacking Beclin1 [337]. This is remarkable evidence of how impaired autophagy may contribute to increased neuroinflammation in AD.

Autophagy can reduce neuroinflammation through degradation of inflammasome and subsequent suppression of IL-1β secretion [338]. IL-1β is a key player in the activation of microglia [339]. Harris et al. reported that autophagy also induces the lysosomal degradation of pro-IL-1β peptide [316]. Accordingly, inhibition of autophagy resulted in increased secretion of IL-1β in an NLRP3- and TRIF-dependent manner [316]. Finally, Bussi et al. reported that induction of autophagy leads to a significant reduction in production of IL-1β, IL-6, TNF-α and NO in BV2 microglial cells. Autophagy induction also inhibited LPS-induced p38 and ERK1/2 phosphorylation in a p38 MAPK and NO-dependent manner [340].

### 6.2. Role of Neuroinflammation in Impairment of Autophagy

The notion of impaired autophagy in AD is against the fact that inflammation in general results in activation of autophagy in cells. Among the increased proinflammatory factors in AD, HMGB1 [341], S100A8/A9 [342], complement receptor [343], Fc receptors [344], scavenger receptors [345], RAGE [346], TLRs [258], and CXCR4 [347] have all been reported to induce autophagy in the cells. Thus, it seems that other factors contribute to the regulation of autophagy in AD.

Thus far, several studies have been conducted trying to find how autophagy is regulated in AD. TLR4 is suggested as one of the main mediators of neuronal damage in AD [348]. Qin et al. reported that stimulation of TLR4 by LPS in tau-transgenic mice resulted in enhancement of neuronal autophagy, which was associated with a reduction of cerebral p-Tau proteins and improved cognitive function [349]. TLR2 has been reported as the receptor for Aβ peptides mediating inflammatory response in microglia upon exposure to Aβ [350]. Arroyo et al. reported that activation of microglial TLR2 using peptidoglycan (PGN) enhances autophagy and results in autophagy-dependent cell death of these cells [351]. Furthermore, François et al. indicated a positive correlation between inflammatory mediators (IL-1β, TNF-α and IL-6) and the expression level of the main autophagy regulators (mTOR and Beclin-1) in the brain of mice with mutations of *App* and *Psen1* genes [352]. All of these studies interpreted that AD and neuroinflammation feed neuro-protective autophagic mechanisms, but their findings just added to the level of controversy [325]. One explanation could be the fact that all of these studies used humanized mouse models of genetic forms of AD hence might not represent the majority sporadic AD cases in human. Interestingly, it is shown that *TREM2* [242] and *APOE4* [353,354] (two genes associated with sporadic AD) negatively regulate autophagy in cells. Additionally, it has been reported that NLRP3 negatively affects autophagy flux through downregulation of PINK1 leading to accumulation of misfolded proteins in AD [318,320,355,356].

## 7. Mitophagy and Neuroinflammation in AD

### 7.1. Role of Impaired Mitophagy in Inflammation

It is believed that mitochondria were incorporated about 1.5 billion years ago through the endosymbiosis of an α-protobacterium in eukaryotic cells [357]. Mitochondria still contain several molecular features from their bacterial ancestors. Moreover, if not removed through the function of mitophagy, these molecules may release from damaged mitochondria and serve as pathogen-associated molecular patterns (PAMPs) and activate pattern recognition receptors (PRRs) [358].

Similar to bacteria, mitochondria have circular DNA harboring unmethylated CpG elements. It has been shown that mtDNA interacts with Cyclic GMP-AMP Synthase (cGAS) and NLRP3 (two important proinflammatory molecules inside the cells) and with Toll-like receptor 9 (TLR-9) when released in to the intracellular space [359,360]. In accordance with this, injection of mtDNA (but not genomic DNA) resulted in tissue inflammation and injury in several animal models [359,361].

Under conditions with excessive generation of ROS, several mitochondrial components, including mtDNA, are oxidized. Excessive ROS also results in the opening of the mitochondrial transition pores (MPTP) mediating the leaking of oxidized mtDNA into the cytoplasm. Within the cells, oxidized mtDNA is a potent proinflammatory signal and results in production of interleukin-1 beta (IL-1β) and IL-18 [362]. Depletion of mtDNA in macrophages (e.g., by cyclosporine) suppressed the production of IL-1β and IL-18 by these cells [362]. It has been shown that the inhibition of autophagy in macrophages increases the accumulation of cytosolic mtDNA in response to lipopolysaccharide (LPS) and ATP. On the other hand, enhancing mitophagy diminished the levels of cytoplasmic mtDNA and inhibited the subsequent activation of the inflammasome [363]. Other mitochondrial derived molecules, which can act as DAMPs, are Adenosine Triphosphate (ATP), Cardiolipin, and Formyl-Peptides [364,365,366,367].

Adding to the level of complexity, mitochondria themselves also contribute to the regulation of danger responses inside the cells and acts as a platform for the transduction of inflammatory signals [358]. Intracellular viral RNAs are detected by two cytosolic proteins Retinoic Acid- Inducible Gene I (RIG-I)-Like Receptors (RLRs) and melanoma differentiation-associated protein 5 (MDA5). Mitochondrial Antiviral Signaling Protein (MAVS) is located in the outer membrane and acts as an adaptor for activated RLRs and relays signals that lead into the activation of key inflammatory genes, such as *NF-κB*, *IFN-I*, and *IFN-III* in the cells [368]. It has been shown that the inhibition of mitophagy enhances MAVS-mediated inflammatory responses through increased production of mtROS and consequent lipid peroxidation of mitochondrial outer membranes [369,370]. Since IFN-I, itself, is a potent enhancer of mtROS, the RLRs/MAVS pathway is thought to be involved in conditions with chronic sustained inflammation [371]. Strikingly, it has been shown recently that IFN-1 is a key driving factor for neuroinflammation and synaptic loss in AD [372].

Likewise, mitochondria also contribute to the regulation of inflammatory responses against exogenous DNA (e.g., from DNA viruses). Cytosolic DNA molecules are detected by the Cyclic GMP-AMP Synthase (cGAS) enzyme. Upon binding to DNA, cGAS increases the levels of cyclic GMP-AMP, which acts as a secondary messenger for Stimulator of Interferon Genes (STING). Under normal conditions, STING is located on mitochondria-associated ER membranes (MAMs) in association with MAVS. After activation, STING translocates to Golgi and perinuclear endosomes and mediates the activation of IRF-3 and NF-κB in the cells [373]. It has been suggested that association with mitochondria might negatively regulate the activity of STING through mediating its ubiquitination and degradation [374].

### 7.2. Mitophagy and Microglia in AD

As mentioned before, accumulation of damaged mitochondria contributes to inflammation within the cells through release of ROS and oxidized biomaterials [84]. Thus, it is believed that the enhancement of mitophagy may reverse neuroinflammation in AD by the removal of damaged mitochondria [375,376,377]. Indeed, it has been reported that mitophagy is impaired severely in microglia isolated from AD brains [118]. In a recent study, Fang et al. showed that enhancement of mitophagy inhibits Aβ and p-tau pathology, and reverses cognitive deficits in an animal model of AD [84]. Accordingly, it was reported that mitochonic acid 5 (MA-5, an inducer of mitophagy) protects mouse microglia against TNFα-induced apoptosis through activation of MAPK–ERK–Yap pathways [378]. This protective effect was mediated by the function of BCL2/adenovirus E1B 19-kDa protein-interacting protein 3 (Bnip3) [378]. Upon induction of mitophagy, Bnip3 translocates to the outer membrane of mitochondria and acts as a mitophagy receptor [379], possibly interacting with LC3 to facilitate fusion of damaged mitochondria with lysosomes [380,381].

## 8. Concluding Remarks

After more than a century of the first description of AD, the real cause(s) of this disorder is (are) still unknown. The amyloid hypothesis has been the dominant theory for AD pathogenesis for more than three decades, but all the clinical trials targeting Aβ have so far failed or show only modest efficacy [382]. Currently, the role of neuroinflammation in AD is under extensive investigation and it has been shown that Aβ is a strong proinflammatory mediator in the brain. As the main brain immune cells, microglia play an important role in neuroinflammation. One of the most striking findings in AD is impaired autophagy. Recent studies on microglia isolated from AD brain confirmed that autophagy is also severely impaired in these cells. Since the crosstalk between autophagy and inflammation is important to prevent noxious effects of inflammation in cells, it is quite plausible to consider that lack of an effective autophagy process in microglia contributes to an aberrant sustained inflammatory state as observed in AD. Interestingly, it seems that the induction of autophagy and mitophagy might show some beneficial effects in AD [190].

Although extensive impairments of autophagy are reported in AD, the exact nature of these pathologies is still unclear. For example, activation of inflammatory pathways in general results in upregulation of autophagy flux in cells as a downstream event. However, reduction in autophagy flux reported does not parallel the neuroinflammatory state reported in the AD brain [383]. Thus, further comprehensive studies are required to find exactly which components of autophagy and mitophagy pathways are affected in AD, as well as how the impairments of autophagy flux affect the inflammatory processes in the AD brain. Specially, it is very important to decipher the crosstalk of autophagy and inflammation within microglia in an attempt to discover more diagnostic biomarkers and therapeutic options in AD.

## Figures and Tables

**Figure 1 ijms-22-03330-f001:**
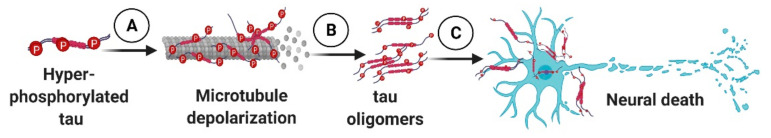
Hyperphosphorylated tau. (**A**) Hyperphosphorylation of tau protein causes microtubule depolarization. (**B**) Tau oligomer aggregation and formation, which assembled to manufacture of neurofibrillary tangles. (**C**) Neural death and release of tau oligomers to the extra-cellular matrix.

**Figure 2 ijms-22-03330-f002:**
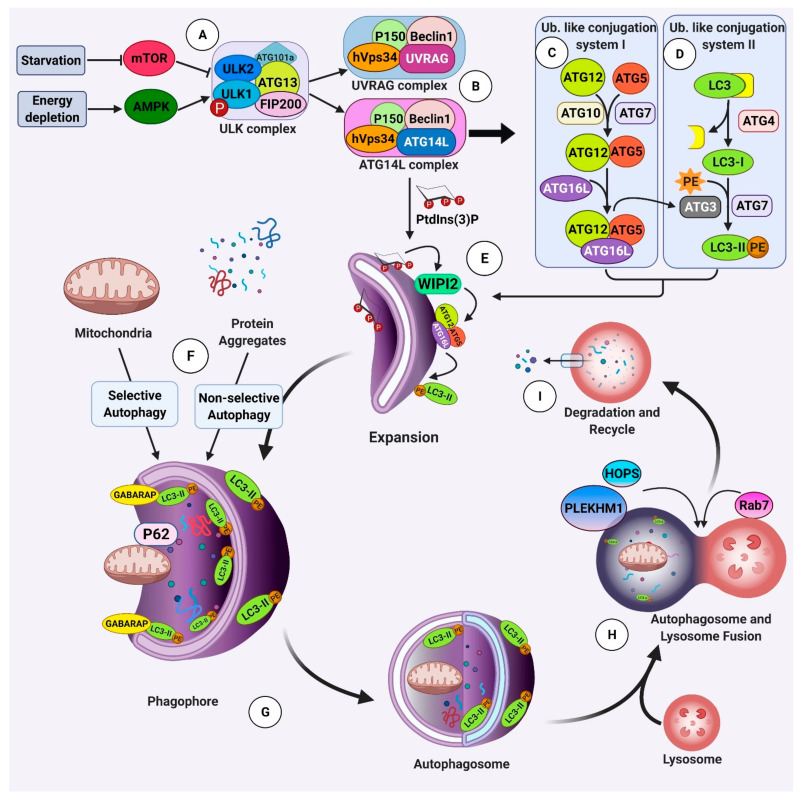
Brief molecular mechanisms of macro autophagy. Autophagy is highly regulated by cascades of multi-molecular complexes consisting of ATG proteins. (**A**) In mammalian cells under physiological conditions, the master switch, mTOR, hinders autophagy by inhibiting the autophagy initiation complex, ULK complex, which consist of ULK1/2, ATG13, FIP200, and ATG 101a. During starvation, because of mTOR inhibition, the ULK complex is activated, triggering autophagy initiation. Additionally, energy depletion activates AMPK, leading to ULK1 phosphorylation, which promotes the assembly of the ULK complex. (**B**) ULK complex activates PtdIn3K complexes including UVRAG and ATG14L complexes. UVRAG complex is involved in autophagosome and lysosome fusion. ATG14L, consisting of Beclin1, hVps34, P150, and ATG14L, triggers the PtdIns(3)P generation required for nucleation and expansion of the phagophore. The autophagy process is followed by the activation of two ubiquitin-like conjugation systems, ATG5-ATG12 conjugation, and LC3 lipidation. (**C**) In the first conjugation system, integration of ATG5 and ATG12 is accrued with the help of ATG7 and ATG10. Then, ATG16L is added to the complex, forming ATG5-ATG12-ATG16L conjugate. (**D**) In the second system, LC3 is cleaved at the C terminus by Atg4, generating LC3-I. Then, LC3-I is conjugated to phosphatidylethanolamine (PE) by a function of ATG7 and ATG3. (**E**) ATG5-ATG12-ATG16L complex localize on the phagophore surface by the help of PtdIns(3)P binding proteins, such as WIPI2. This triggers ATG3 activation and LC3 lipidation. (**F**) The autophagy targets, such as protein aggregates and cytoplasmic components, can be non-selectively sequestrated within the double membrane contract, autophagosome. Selective autophagy targets specific cellular components for degradation using different adaptor proteins, such as P62. (**G**) Autophagosome is formed upon completion and closure of the phagophore membrane by the help of LC3 and GABARAP proteins. (**H**) Autophagosome, the completed phagophore, is then fused with lysosome with the help of other proteins such as Rab7 on the lysosome and HOPS and PLEKHM1 on the autophagosome surface. (**I**) Subsequently, targets of autophagy are degraded by lysosomal hydrolases, and their building blocks are recycled back into the cytosol.

**Figure 3 ijms-22-03330-f003:**
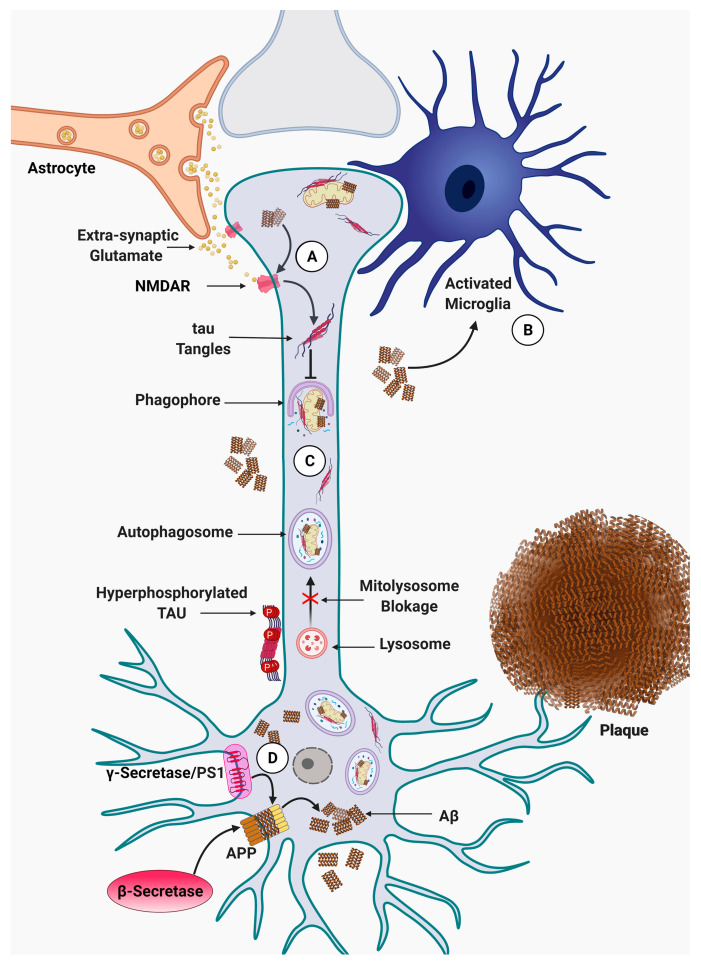
Mitophagy and microglia activation in Alzheimer disease. (**A**) Increased release of extra-synaptic glutamate from astrocytes, which results in extra-synaptic N-methyl-D-aspartate (NMDA) receptors activation, accompanied by the interference of Aβ oligomers with those receptors and inhibition of pro-survival pathways, triggering the pro-apoptotic signaling pathway. This phenomenon leads to dendritic spine loss and postsynaptic depression. (**B**) Microglia activation, which is induced by Aβ oligomers, proceeds inflammation and neurodegeneration via interference with immunological processes in the brain. (**C**) Impaired mitophagy represented as phagophore and mitolysosme blockage is promoted by p-tau, which leads to an energy production decline and increased oxidative stress, (**D**), which causes APP processing by β secretase and γ secretase/PS1. Accumulation of Aβ and P-Tau participate in mitophagy impairment, which initiates a vicious cycle.

**Figure 4 ijms-22-03330-f004:**
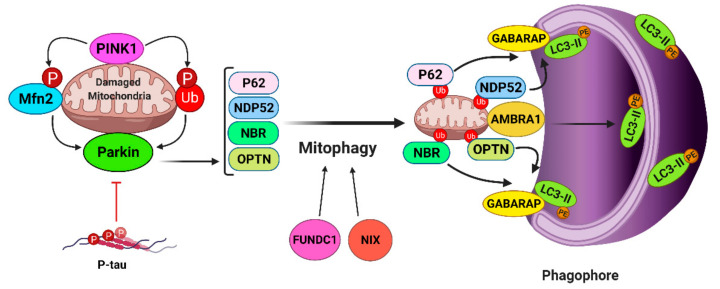
Mitophagy; mitochondria targeting. During mitochondrial dysfunction, PINK1 is localized on the outer mitochondrial membrane. PINK1 activates Mfn2 as well as Ubiquitin molecules by phosphorylation, leading to the recruitment of Parkin an E3 ubiquitin ligase. Then, Ubiquitin binding mitophagy receptors such as P62, NDP52, NBR, and OPTN act as adaptor proteins and direct mitochondria into the expanding phagophore through covalently binding to LC3 and GABARAP. While hyperphosphorylated tau (p-tau) is found to block mitophagy by inhibiting Parkin, NIX and FUNDC1 proteins are inducers of mitophagy. AMBRA1 can also promote mitophagy by binding to LC3.

**Figure 5 ijms-22-03330-f005:**
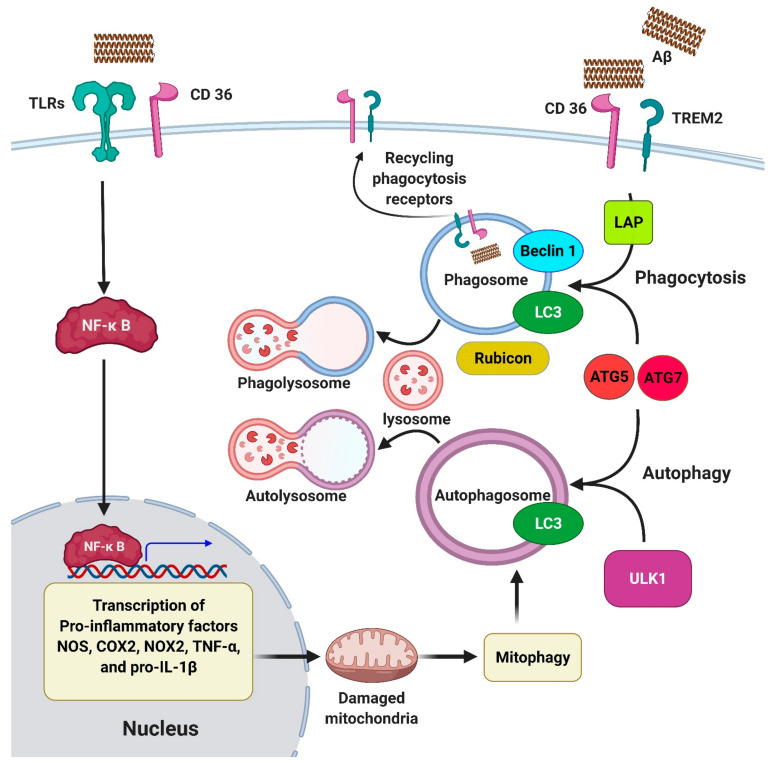
Cooperation of autophagy with phagocytosis in microglial activation. Autophagy and phagocytosis are tightly intertwined, making them promising therapeutic targets in Alzheimer’s diseases. Autophagy could be activated by activation of ULK 1. Interaction of Aβ with TREM2 and CD36 at the surface of microglia initiates phagocytosis, which is facilitated by autophagy mediators. LC3 recruitment to the single-membrane phagosomes is mediated by the activity of ATG 5 and ATG 7. Thereafter, Beclin1 and LC3 proteins translocate to the phagosomes, enhancing the fusion of phagosome into the lysosomal system. The interaction of autophagy and phagocytosis, through LC3-associated phagocytosis (LAP), enhances the efficiency of microglial phagocytosis in the elimination of extracellular cargos including Aβ aggregates. Rubicon’s activity is critical in the production of PtdIns(3)P at the phagosome, triggering the recruitment of the ATG5–12 and LC3-PE conjugation systems. On the other hand, toll-like receptors (TLRs) are involved in the induction of microglial autophagy. TLRs initiate signaling pathways, such as NF-κB. This pathway triggers the transcription of pro-inflammatory factors, such as NOS, COX2, NOX2, TNF-α, and pro-IL-1β, which are damaging to mitochondria, and consequently, induce the autophagic flux.

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
