# Peer review of "Alzheimer’s Disease Pathogenesis: Role of Autophagy and Mitophagy Focusing in Microglia"

_ijms, 2021, doi:10.3390/ijms22073330_

Round 1
Reviewer 1 Report
The authors provide a detailed review on the interrelated mechanisms of neuroinflammation and impaired autophagy/mitophagy in AD. The review is of interest, has merit. In this Reviewer opinion, the following major concerns must be adressed. In addition the text needs revision of English style, a number of such editorial remarks have been listed to aid that process.
Major concerns:
- Figure 2 is very difficult to decipher. For instance, it opens with „extrasynaptic glutamate” – while the figure shows a synapse. There is no indication how glutamate is involved in subsequent processes. In general, the sequence of events is not explained by the Figure legend, and the message remains obscure.
- Figure 4 in its present form suggests that microglial mitochondria are damaged and destructed in AD pathology, assuming that the CD36+ cell is a microglial cell. Is this the intention of the authors? Please explain.
- In the concluding remarks, a whole new line of thought starts (line 864 Secretory autophagy). This text should be not in the conclusion.
Minor content remarks:
Line 77 50 million people is 5% of 1 billion, and 8% of somewhat less people. As the global population is much larger than 1 billion, this statement cannot be true in the present form. Please correct.
Line 147 „genes interact” – probably what is meant that the tranlated proteins, the gene products interact. Please correct the sentence to reflect this meaning.
Line 165 mTOR is the same as mTORC1 in text?
Line 216 „iPS” is not explained in text
Line 273 – Figure 1 does not show AMBRA1 although the text refers to it here
Line 278 – Figure 1 does not show GABARAP although the text refers to it here
Line 563 „…plays a role in clearance of AD pathology…” What does this meand Please rephrase statement.
Line 691 no H part of Figure 3!
Style remarks:
Line 38 „mitochondria phagy” can be omitted
Line 131 „causal” is meant rather than „casual”, right?
Line 146 „mammals” is recommended to use instead of „mammalians”
Line 154 „homologue” instead of „homolog”
Line 161 „are” instead of „is” is correct grammatically
Line 175 a comma would be perhaps best inserted after „targets” - …targets, such as ….
Line 187 „Figure 1” bold font – but not throughout the manuscript (eg Line 273), also for Figure 2 (Lines 362, 364)
Line 233 „alleviated” reads better than „alleviates”
Line 287, 685 „Alzheimer” instead of „Alzhiemer”
Line 289 „energy” does not need capital „E”
Line 289 „increased oxidative stress” reads better than „oxidative stress increased”
Lines 305-315 – font color is not black but gray?
Lines 337-338, 471, 506, 508, 514, 543-45, – font size is larger than rest of text?
Line 405, 855 „beneficial” reads better than „beneficiary”
Line 406-408 more conventional scientific style is advisable instead of colloquial
Line 493 „Microglial.” – Missing sentence?
Line 574 „by the all cell types” – the „the” should be deleted?
Line 615 long gap between „S100” and „is” – perhaps something is missing
Line 640 „Post-mortem” is recommended instead of „postmortem”
Line 655 „machinery” instead of „mavhinery”
Line 683 „details” or „detailed explanation” instead of „detailed”
Line 687 … senile neurons that initiate… reads better
Line 692 „induced” instead of „inudced”
Line 736 „Beclin1‑driven” instead of „Beclin1driven” is meant?
Line 839 Either Bnip3 or BNIP3 should be used as an abbreviation.
Line 866 „gets released” reads better than „released”
Line 870 … cytokines and the secretion… reads better („and” is missing)
Line 873 „crosstalk” instead of „cross talk”
Line 875 omit „disease” and use comma after „on the other hand,”
Author Response
Reviewer #1
The authors provide a detailed review on the interrelated mechanisms of neuroinflammation and impaired autophagy/mitophagy in AD. The review is of interest, has merit. In this Reviewer opinion, the following major concerns must be addressed. In addition, the text needs revision of English style, a number of such editorial remarks have been listed to aid that process.
A: We truly appreciate the reviewer’s editorial remarks. We have changed the mentioned points accordingly. Also, the text was revised to provide a better English style.
Major concerns:
Figure 2 is very difficult to decipher. For instance, it opens with „extrasynaptic glutamate” – while the figure shows a synapse. There is no indication how glutamate is involved in subsequent processes. In general, the sequence of events is not explained by the Figure legend, and the message remains obscure.
A: To insure a better understanding about the role of hyper-phosphorylated tau in the pathogenesis of AD one figure (now Figure 1) is added to the manuscript.
Also the old Figure 2 (now Figure 3) is simplified for a better representation of pathologic alterations of mitophagy in AD. The figure now clearly shows how “extrasynaptic glutamate” initiates the pathologic events in AD. The Figure legend was also extensively revised for a better explanation of the sequence of pathologic events in AD.
Figure 4 in its present form suggests that microglial mitochondria are damaged and destructed in AD pathology, assuming that the CD36+ cell is a microglial cell. Is this the intention of the authors? Please explain.
A: Thanks for this valid point. The legend of the old Figure 4 (now Figure 5) is revised extensively for a clear explanation of the represented cell (i.e. microglia) and sequence of pathologic events in microglia in AD.
In the concluding remarks, a whole new line of thought starts (line 864 Secretory autophagy). This text should be not in the conclusion.
A: The mentioned paragraph was removed from the manuscript.
Minor content remarks:
Line 77 50 million people is 5% of 1 billion, and 8% of somewhat less people. As the global population is much larger than 1 billion, this statement cannot be true in the present form. Please correct.
A: Line 77: A valid point. The percentage was removed from the text.
Line 147 „genes interact” – probably what is meant that the translated proteins, the gene products interact. Please correct the sentence to reflect this meaning.
A: Now Line 157: A valid point. The text is corrected accordingly.
Line 165 mTOR is the same as mTORC1 in text?
A: mTOR is the core protein of two protein complexes in the cells (i.e. mTORC1 and mTORC2). Here we wanted to emphasize the role of mTORC1 in inhibition of autophagy.
Line 216 „iPS” is not explained in text
A: Now Line 227: The text is modified for a clearer statement.
Line 273 – Figure 1 does not show AMBRA1 although the text refers to it here
A: AMBARA1 is one of the several receptors of autophagy as discussed in the manuscript. To avoid the complexity, we only showed one of the autophagy receptors (p62) in the old Figure 1 (now Figure 2).
Line 278 – Figure 1 does not show GABARAP although the text refers to it here
A: The old Figure 1 (now Figure 2) is revised accordingly.
Line 563 „…plays a role in clearance of AD pathology…” What does this mean Please rephrase statement.
A: Now Line 595: Thanks for the valid point. The word clearance is removed from the text. And the rest of the paragraph explains how CX3CL1 plays an important role in AD pathology.
Line 691 no H part of Figure 3!
A: The legends of all figure are revised extensively and mentioned errors are now corrected.
Style remarks:
Line 38 „mitochondria phagy” can be omitted
A: The text is revised accordingly.
Line 131 „causal” is meant rather than „casual”, right?
A: Now line 141: A valid point. The text is revised to be clearer.
Line 146 „mammals” is recommended to use instead of „mammalians”
A: Now Line 156: The text is corrected.
Line 154 „homologue” instead of „homolog”
A: Now Line 164: The text is corrected.
Line 161 „are” instead of „is” is correct grammatically
A: Now line 171: The text is corrected.
Line 175 a comma would be perhaps best inserted after „targets” - …targets, such as ….
A: Now line 191: The text is corrected.
Line 187 „Figure 1” bold font – but not throughout the manuscript (eg Line 273), also for Figure 2 (Lines 362, 364)
A: These typo errors are now corrected.
Line 233 „alleviated” reads better than „alleviates”
A: Now Line 243: the text is revised accordingly.
Line 287, 685 „Alzheimer” instead of „Alzhiemer”
A: These typo errors are now corrected.
Line 289 „energy” does not need capital „E”
A: This typo error is corrected.
Line 289 „increased oxidative stress” reads better than „oxidative stress increased”
A: Now Line 266: the text is corrected accordingly.
Lines 305-315 – font color is not black but gray?
A: The font color is changed to black.
Lines 337-338, 471, 506, 508, 514, 543-45,– font size is larger than rest of text?
A: The font size of the text is now corrected.
Line 405, 855 „beneficial” reads better than „beneficiary”
A: Now line 442 and 884: The text is revised accordingly.
Line 406-408 more conventional scientific style is advisable instead of colloquial
A: The paragraph is revised extensively to make a scientific statement.
Line 493 „Microglial.” – Missing sentence?
A: Now Line 529: thanks. This typo error is corrected.
Line 574 „by the all cell types” – the „the” should be deleted?
A: Now Line 610: “the” was removed.
Line 615 long gap between „S100” and „is” – perhaps something is missing
A: The text is corrected accordingly.
Line 640 „Post-mortem” is recommended instead of „postmortem”
A: Line 678: the text is corrected accordingly.
Line 655 „machinery” instead of „mavhinery”
A: The text is corrected accordingly.
Line 683 „details” or „detailed explanation” instead of „detailed”
A: Line 721: “detailed explanation” is used now.
Line 687 … senile neurons that initiate… reads better
A: The text is corrected accordingly.
Line 692 „induced” instead of „inudced”
A: The text is corrected accordingly.
Line 736 „Beclin1‑driven” instead of „Beclin1driven” is meant?
A: Line 770: is corrected now.
Line 839 Either Bnip3 or BNIP3 should be used as an abbreviation.
A: Line 873: Bnip3 is used instead of BNIP3.
Line 866 „gets released” reads better than „released”
A: Line 901: the whole paragraph is removed based on a previous major comment.
Line 870 … cytokines and the secretion… reads better („and” is missing)
A: Line 905: the whole paragraph is removed based on a previous major comment.
Line 873 „crosstalk” instead of „cross talk”
A: Line 908: the typo error is corrected.
Line 875 omit „disease” and use comma after „on the other hand,”
A: Line 910: the text is corrected accordingly.
Reviewer 2 Report
General comment:
In the manuscript entitled " Alzheimer’s Disease Pathogenesis: Role of Autophagy and Mitophagy Focusing in Microglia” Eshraghi and colleagues present a review of the literature on several topics linked to the neurodegenerative Alzheimer disease. After a short introduction (chapter 1) and a global health positioning of the topic (chapter 2), the third and four parts are dedicated to autophagy and mitophagy in AD, respectively. Chapter 5 is a long part is focussing in microglia and neuroinflammation in AD and the chapters 6 and 7 aim to present the links between auto/mitophagy and neuroinflammation in AD, before a conclusive paragraph.
The review is well written and individual chapters contain interesting data with a long list of references.
Major modifications:
The major remarks concern the organization of the paper and the figures.
In §3.1 references are often not pertinent 1 where the authors should cite the original and often seminal papers in the field of autophagy, rather that reviews.
The links between autophagy/mitophagy and AD in Figure 2 is not clear (mechanisms or correlation). The legend should be rewritten and further detailed.
All the chapter 5 is informative but the relation with the topic and the tittle of the review is not obvious.
Most of the data in chapter 7 are not really demonstrating a strong link between inflammation and mitophagy, because no details concerning mitophagy experiments are given.
Several illustrative aspects of the figures are very redundant and some tables would be much more informative
Minor modification.
“mitochondria phagy” is an inappropriate term (line 38)
Alzhiemer instead of Alzheimer (fig 2- line287 and fig3 line685).
Lines 150-1, the so-called LC3-ATG7 complex should not be really considered as a bona fide autophagic complex but rather a transitory step during the post translational modification of LC3, with the conjugation systems.
Lines 337-8 check size and police.
Fig 3, the relation between bottom of the figure and autophagy is unclear - H is not explained
Author Response
Reviewer #2
General comment:
In the manuscript entitled " Alzheimer’s Disease Pathogenesis: Role of Autophagy and Mitophagy Focusing in Microglia” Eshraghi and colleagues present a review of the literature on several topics linked to the neurodegenerative Alzheimer disease. After a short introduction (chapter 1) and a global health positioning of the topic (chapter 2), the third and four parts are dedicated to autophagy and mitophagy in AD, respectively. Chapter 5 is a long part is focussing in microglia and neuroinflammation in AD and the chapters 6 and 7 aim to present the links between auto/mitophagy and neuroinflammation in AD, before a conclusive paragraph.
The review is well written and individual chapters contain interesting data with a long list of references.
A: We appreciate the nice comments by the reviewer #2. We believe that the role of impaired autophagy and mitophagy is an overlooked issue in AD. Also, impairments of autophagy/mitophagy in microglia is a very recent topic and not explored deeply. So, our main goal of the current manuscript is to cover the current knowledge about impairments of autophagy and neuroinflammation in AD. We are also trying to persuade the scientific society for the further exploration of crosstalk between these two pathologies in AD as possible therapeutic targets.
Major modifications:
The major remarks concern the organization of the paper and the figures.
In §3.1 references are often not pertinent 1 where the authors should cite the original and often seminal papers in the field of autophagy, rather that reviews.
A: We have followed the respected reviewer comment and made the following changes:
Ref # 58 referred at line 141
.........71.........................155
..........73.........................161
..........74.........................161
..........75.........................161
..........76.........................163
..........78.........................170
..........82.........................205
..........83.........................208
The links between autophagy/mitophagy and AD in Figure 2 is not clear (mechanisms or correlation). The legend should be rewritten and further detailed.
A: To insure a better understanding regarding the role of hyperphosphorylated tau in the pathogenesis of AD one figure (now Figure 1) was added to the manuscript.
Also the old Figure 2 (now Figure 3) is simplified for a better representation of pathologic alterations of mitophagy in AD. The figure now clearly shows how “extrasynaptic glutamate” initiates the pathologic events in AD. The Figure legend was also extensively revised for a better explanation of the sequence of pathologic events.
All the chapter 5 is informative but the relation with the topic and the tittle of the review is not obvious.
A: Based on the scope of the paper, we felt obligated to cover the details of microglial over-activation in the pathogenesis of AD. In addition, considering the emerging knowledge regarding the role of different molecular pathways in microglia in AD, we wanted to discuss the possible crosstalk between each molecular pathway and autophagy in the later chapters of the manuscript.
Most of the data in chapter 7 are not really demonstrating a strong link between inflammation and mitophagy, because no details concerning mitophagy experiments are given.
A: In Chapter 7, we tried to put together all the current knowledge of crosstalk between mitophagy and inflammation in AD. In section 7.1 we provided evidence and current opinions of how defective mitochondria contributes to aberrant inflammation in general. In section 7.2, we specifically discussed the consequences of defective mitophagy on microglia in AD. We agree that there is not enough knowledge about how defective mitophagy mediates over-activation of microglia in AD and therefore would like to encourage the scientific society to pay more attention to this topic.
Several illustrative aspects of the figures are very redundant and some tables would be much more informative.
A: All the figures and their legends of the manuscript are revised extensively. The figures are now more simple to understand. The legends also include more details. We also made sure to remove the redundant part of the figures.
Minor modification:
“mitochondria phagy” is an inappropriate term (line 38)
A : The text is revised accordingly.
Alzhiemer instead of Alzheimer (fig 2- line287 and fig3 line685).
A: The text is revised accordingly.
Lines 150-1, the so-called LC3-ATG7 complex should not be really considered as a bona fide autophagic complex but rather a transitory step during the post translational modification of LC3, with the conjugation systems.
A: Thanks for the valid point. The LC3-ATG7 is removed from the text.
Lines 337-8 check size and police.
A: The format is corrected accordingly.
Fig 3, the relation between bottom of the figure and autophagy is unclear - H is not explained
A:The figure and its legend are revised extensively to comply wi
Round 2
Reviewer 1 Report
The authors heeded the critique and modified the manuscript accordingly, redrew and explained Figures. I have no further major concern.
By checking the references, I find Ref #15 and #17 are not identifieable with certainty please cite them according to journal style.